# Relevant Biomarkers in Medical Practices: An Analysis of the Needs Addressed by an International Survey

**DOI:** 10.3390/jpm12010106

**Published:** 2022-01-14

**Authors:** Laure Abensur Vuillaume, Thierry Leichle, Pierrick Le Borgne, Mathieu Grajoszex, Christophe Goetz, Paul L Voss, Abdallah Ougazzaden, Jean-Paul Salvestrini, Marie-Pia d’Ortho

**Affiliations:** 1Emergency Department CHR Metz-Thionville, 57000 Metz, France; 2CNRS, IRL 2958, GT-CNRS, 2 Rue Marconi, LAV, 57070 Metz, France; thierry.leichle@cnrs.fr (T.L.); pvoss@georgiatech-metz.fr (P.L.V.); abdallah.ougazzaden@georgiatech-metz.fr (A.O.); jpsalvestr@georgiatech-metz.fr (J.-P.S.); 3Georgia Tech-Lorraine, 2 Rue Marconi, 57070 Metz, France; 4LAAS-CNRS, 31400 Toulouse, France; 5Emergency Department, Hopitaux Universitaires de Strasbourg, 67000 Strasbourg, France; Pierrick.LEBORGNE@chru-strasbourg.fr; 6INSERM (French National Institute of Health and Medical Research), UMR 1260, Regenerative NanoMedicine (RNM), Federation de Médecine Translationnelle (FMTS), University of Strasbourg, 67000 Strasbourg, France; 7Digital Medical Hub, APHP, 75000 Paris, France; mathieu.grajoszex@aphp.fr (M.G.); marie-pia.dortho@aphp.fr (M.-P.d.); 8Clinical Research Support Unit, CHR Metz-Thionville, 57000 Metz, France; c.goetz@chr-metz-thionville.fr; 9Georgia Institute of Technology, School of Electrical and Computer Engineering, GT-Lorraine, 57070 Metz, France; 10Service de Physiologie Clinique-Explorations Fonctionnelles, Hopital Bichat, APHP, 75018 Paris, France; 11INSERM, Neurodiderot, Universite de Paris, 75019 Paris, France

**Keywords:** biosensors, biomarkers, emergency overcrowding

## Abstract

(1) Backround: Technological advances should foster gains in physicians’ efficiency. For example, a reduction of the medical decision time can be enabled by faster biological tests. The main objective of this study was to collect responses from an international panel of physicians on their needs for biomarkers and also to convey the improvement in the outcome to be made possible by the potential development of fast diagnostic tests for these biomarkers. (2) Methods: we distributed a questionnaire on the Internet to physicians. (3) Results: 508 physicians participated in this survey. The mean age was 38 years. General practice and emergency medicine were heavily represented, with 95% CIs of 44% (39.78, 48.41) and 32% (27.84, 35.94)), respectively. The two most represented countries were France (95% CI: 74% (70.20, 77.83)) and the USA (95% CI: 11% (8.65, 14.18)). Ninety-eight percentages of the physicians thought that obtaining cited biomarkers more quickly would be beneficial to their practice and to patient’s care. The main biomarkers of interest identified by our panel were troponin (95% CI: 51% (46.24, 54.94)), C-reactive protein (95% CI: 42% (38.03, 46.62)), D-dimer (95% CI: 29% (24.80, 32.68)), and brain natriuretic peptide (95% CI: 13% (10.25, 16.13)). (4) Conclusions: Our study highlights the real technological need for fast biomarker results, which could be provided by biosensors. The relevance of some answers such as troponin is questionable.

## 1. Introduction

Even though the SARS-CoV-2 pandemic has highlighted the saturation of healthcare systems, this saturation already existed well before the pandemic hit the world, with a major overcrowding of the emergency departments (EDs) in France and abroad [1,2,3]. The numbers of visits to EDs constantly increased, e.g., +36% in the United States between 1993 and 2003 [4] and +3.5% per year between 1996 and 2018 in France [5,6]. More generally, the pressure on hospital capacity is associated with the increased morbidity and mortality and deterioration of care [7,8,9]. 

Such pressure demands finding ways to make ED care more efficient. One of the possible levers, both in terms of the safety and quality of care, could be to optimize the medical decision time. This time is lengthened by several causes, which include the time to wait for the results of biological biomarker assays [10]. Today, most assays are performed by centralized laboratories within 30 min to 1 h, with a total processing time from the sample delivery to the lab to the communication of the result, sometimes taking up to a day in a city hospital, even if the analysis is conducted in the same city. The technological development of breakthrough technologies such as novel biosensors could have a major impact on reducing delays. However, from the perspective of the development of such sensors, it is absolutely essential to correctly identify the needs of professionals and to identify markers for which the marketing of a biosensor would provide a real advantage over standard techniques, both in terms of analysis time in the field and in terms of at least equal performance and competitive price [11]. 

The main objective of this study was to collect responses from an international panel of physicians regarding their need for biomarkers and also to know the anticipated consequences in terms of health care if such technologies were to be developed. 

## 2. Materials and Methods

### 2.1. Study Design and Patient Population

An electronic survey was conducted among physicians. The study gathered responses from 508 physicians throughout the world. Medical Associations, the Council of the Order of Physicians (e.g., Conseil national de l’ordre des médecins), Regional Unions in Healthcare, and physician groups on social networks were asked to disseminate a proposal to participate in the survey. This questionnaire, after an expert review, was previously validated after testing it on a representative sample of physicians (composed of 10 emergency physicians). We made sure that the questions were understood, as we specifically sought for comments regarding the understanding of the questionnaire. 

The first part of the survey included demographic questions (age, medical specialty, country, and practice pattern); the second part included two questions about biomarkers of interest as well as their relevance for use (questionnaire available in Appendix A). Regarding the need for biomarkers, the physicians were free to suggest any marker that they considered to be relevant. Each response was then manually classified to allow for analysis.

Ethical committee approval was not required for this type of professional survey, because the collected data were anonymous. The participation in the questionnaire served as consent, with the scientific purpose well explained in the preamble.

### 2.2. Statistical Analysis

Qualitative data were described in terms of numbers and percentages, and quantitative data were described in terms of the mean and standard deviation. Respondent characteristics were compared by the major tests cited using Student’s *t*-tests for quantitative data and Fisher’s exact tests for qualitative data, followed by logistic regressions. Statistical analyses were performed with SAS software (version 9.4, SAS Inst., Cary, NC, USA). The significance level was set at 5%.

## 3. Results

### 3.1. Characteristics of Participating Physicians 

A total of 508 physicians participated in this survey between June 2021 and October 2021. There were no missing data. Answers were precise enough to enable interpretation. The mean age was 38 years (min–max: 22–74). General practice and emergency medicine were the most represented, with a 95% CI of 44% (39.78, 48.41) and a 95% CI of 32% (27.84, 35.94), respectively. The two most represented countries were France (a 95% CI of 74% (70.20, 77.83)) and the USA (a 95% CI of 11% (8.65, 14.18)). Most of the physicians were practicing in the public hospital (a 95% CI of 61% (56.78, 65.26)). The demographic and professional characteristics of respondents are summarized in Table 1.

### 3.2. Biomarkers Requirement for Obtaining Fast Results in Daily Practice

The large majority of physicians (a 95% CI of 98% (96.57, 99.10)) believed that obtaining a faster result for the cited biomarkers’ assays would be beneficial to their patients and the quality of care in daily practice. The others did not give any argument to explain their negative answers. The main biomarkers of interest identified by our panel were troponin (a 95% CI of 51% (46.24, 54.94)), C-reactive protein (CRP) (a 95% CI of 42% (38.03, 46.62)), D-dimer (a 95% CI of 29% (24.80, 32.68)), creatinine and urea (a 95% CI of 19% (15.13, 21.88)), blood cell count (a 95% CI of 18% (14.21, 20.83)), and brain natriuretic peptide (BNP) or N-terminal (NT)-proBNP (a 95% CI of 13% (10.25, 16.13)). The list of all biomarkers cited by the panel is provided in Table 2.

Among physicians who found value in their practice, we classified the verbatim of responses into three categories. Thus, physicians mentioned with a 95% CI of 50% (45.26, 53.95) indicated “a time gain, fluidification for hospital or a gain in private practice”, physicians mentioned with a 95% CI of 62% (58.39, 66.81) indicated an improvement as “guiding a course of action, helping to make a therapeutic decision, avoiding useless treatment while waiting for a result, and avoiding diagnostic delay”, and physicians mentioned with a 95% CI of 34% (29.55, 37.77) answered the more rapid referral of patients to an appropriate structure. Physicians with a 95% CI of 4% (2.25, 5.63) only did not express any interest in obtaining a quick result for any biomarker.

### 3.3. Details about the Biomarkers of Interest

We also compared responses of physicians practicing in public hospitals with those practicing in private practice and responses of physicians practicing in France versus abroad. S100B protein, blood gases, creatinine, and procalcitonin were more frequently requested by public practitioners (*p* < 0.005), whereas private practitioners more frequently quote the blood cells count and the CRP. These data are available in Table A1 and Table A2 (Appendix B).

Concerning the markers most frequently cited by the panel, there was less interest in respiratory virology tests in private practice than in countries outside Europe, France, or the USA. Regarding cardiac biomarkers, troponin was of most interest to EDs, followed by general practitioners (OR: 3.58; 95% CI: 1.82–7.03; *p* < 0.001) and then other specialists. As for D-dimer, it was of more interest to emergency rooms and general practitioners than to other specialties (OR: 0.24; 95% CI: 0.11–0.53; *p* < 0.001) and was of less interest to practitioners in the USA than in France. BNP and proBNP were of interest to all physicians, regardless of the characteristics studied (Table 3; all data are in Table A3).

## 4. Discussion

Our survey of an international panel of physicians confirmed that there was a strong need to speed up care decisions and to speed the flow of information by quickly obtaining biology results, in both hospitals and private practice in France and abroad. The results highlighted a certain number of biomarkers of interest. The most requested by our panel were cardiovascular markers (troponin, BNP or NT-proBNP, and D-dimers), in connection with major cardiac emergencies (infarction, cardiac respiratory distress, and pulmonary embolism) and the fact that a biological result can influence the management and orientation of a patient by excluding a diagnosis due to its negative predictive value. Finally, benefits in terms of rapid referral, the quality and safety of care and medico-economic impact were widely emphasized by our panel.

Our results also pointed out the possibility of a gap between the needs expressed by our panel of physicians and the current medical guidelines. For example, the rapid access to a troponin result may be controversial, at least in the emergency and clinical setting of chest pain. Indeed, troponin is not useful if the infarction is clinically evident or in symptomatic patients without obvious electrical changes, but with a very high vascular risk. It is, however, useful to exclude or to confirm the diagnosis via a troponin assay in cases where the pain is not associated with an electrocardiogram (ECG) change in patients at medium or low risk who may have a true subclinical infarction [12,13]. In less typical cases, it is necessary to evaluate the kinetics between two samples [13]. If the kinetics is positive (i.e., when there is a 20% increase between two samples), revascularization must take place within 24 h; the patient is then either hospitalized or discharged, following a stay in the emergency room of about four hours. Thus, a rapid result in the face of chest pain is of little benefit to the patient or his safety if he is in the emergency room, since he is monitored and his immediate management is thus not modified. The time of waiting for the second test result could be reduced, but the balance between the time gained and the development cost is far from positive. On the other hand, the question of a rapid test may be relevant in pre-hospital medicine, in the ED, or in a general practitioner’s or cardiologist’s office [10,14]. Indeed, in this context of acute chest pain, the performance of a conventional biological test (outside of the hospital), if positive, can represent a real loss of chance for the patient. If it is negative, it does not formally allow the elimination of an ischemic phenomenon since it is the kinetics that provides a useful answer. On the other hand, a rapid test to obtain D-dimer levels in cases of clinical suspicion of thrombosis or pulmonary embolism would be an interesting tool [15,16,17], since D-dimer has an excellent negative predictive value for the diagnosis of thrombosis (that is ruled out in the event of a normal value).

Today, biological tests present temporal problems that are not conducive to accelerated management or referral decisions. Nevertheless, biological tests in central laboratories are now the gold standard in all good practice recommendations. From the point of view of acceleration, carrying out an analysis at the patient’s bedside allows for a reduction in the time required for the transport and validation of results. A certain number of rapid diagnostic tests such as direct reading tests on immunochromatographic-type techniques are developed in this sense, particularly in infectiology [18,19]. These tests, which can be read directly or sometimes automatically by a computer, are not as reliable as the gold standard and are most often used for orientation without confirmation (qualitative tests, most of the time) [20]. On the other hand, point-of-care tests (POCTs), which are often larger and less portable, are much more reliable. This equipment is found in many hospitals, particularly in emergency rooms. However, the drawbacks of this technology include the fact that it remains under the full responsibility of a biologist, and its installation and use is costly. The associated cost of the technology is inversely proportional to the volume of use, and simulated economic impact studies have shown that routine use would reduce the overall cost to society [21,22]. Still, the initial investment cost is probably the main obstacle that prevents medium-sized hospitals (and even less so community medicine) investing in the POCT technology. However, these technologies are widely used in large centers. 

It should be noted that in this field of rapid tests, biosensors derived from microelectronics have received increasing attention over the last decade, particularly in the field of POCT, and could be used for a large number of medical applications [23]. It is important to consider all the characteristics of biomarkers that one wishes to detect with a rapid test, especially using new technologies aiming at competing against or complementing gold standards, i.e., its sensitivity, specificity, positivity threshold, kinetics, sampling method, type of sample, cost of analysis, and, last but not least, its usefulness in clinical practice. Take troponin as an example. A great deal of basic engineering research is focused on developing sensors for troponin detection; however, as previously discussed, it is more than likely that this target is not the most relevant [24,25]. The benefit for patient management must come first in order for efficiency gains to outweigh the costs of testing. Apart from the initial cost of the basic research, this type of technology could be of low cost to the clinician, while having the same reduction in the overall societal cost as POCTs [22]. These technologies must be highly adaptable and probably address the possibilities of multimarkers approaches to meet the needs of physicians, such as a multimarker tool for cardiac biomarkers or infectious biomarkers [26,27]. These approaches, beyond the practical interest of obtaining several concomitant results with the same technique, would make it possible to implement the notion of gravity scores and the weighting of each marker.

### Limitations

Our study has a number of limitations. Indeed, the sample of physicians could have been more representative of the world population; our study was distributed from France, which induced a recruitment bias. Nevertheless, our study presents a global coherence with other work found in the literature [25,28,29,30], which makes it an interesting and relevant basis for future research in the field of new technologies.

## 5. Conclusions

Our study showed physicians had a strong interest in obtaining more rapid biomarker results such as troponin, D-dimer, or BNP. The potential time-savings associated with the use of a technology, such as biosensors, which provides access to very rapid biomarker results identified by our panel of physicians, would allow for greater safety of care and would improve the triage orientation and treatment decisions of patients not only in the hospital, but also in the city. It is important to note, however, that the relevance of faster diagnostics for some of the cited biomarkers, such as troponin, is questionable from the standpoint of clinical utility, raising questions about the knowledge of medical guidelines in the physician community.

## Figures and Tables

**Table 1 jpm-12-00106-t001:** Characteristics of the participating physicians.

Characteristics		*n*	Mean ± SD/*n* (%)	Min–Max
Age (years)		508	38 ± 10	22–74
Status		508		
	Medical doctor		407 (80)	
	Resident		101 (20)	
Speciality		508		
	Emergency physician		224 (44)	
	General practice		162 (32)	
	Cardiologist		20 (4)	
	Other		27 (5)	
Location		508		
	France		376 (74)	
	USA		58 (11)	
	Europe (excluding France)		47 (9)	
	Other		27 (5)	
Activity		508		
	Public practice		310 (61)	
	Private or mixed practice		198 (39)	

**Table 2 jpm-12-00106-t002:** Respondent specifications of biomarkers needed for obtaining fast results in physicians’ daily practice.

	*n*	Outcome N (%)
Favorable physicians	508	497 (98)
Cited biomarkers		
Troponin		257 (51)
CRP		215 (42)
D-dimer		146 (29)
Creatinine and urea		94 (19)
Blood cell count		89 (18)
BNP or N-terminal (NT)-proBNP		67 (13)
Ionogramme		62 (12)
hCG		49 (10)
Blood gas and lactate		46 (9)
Procalcitonin		41 (8)
Respiratory virus test		36 (7)
Hemostase		25 (5)
Glycemia		19 (4)
Hepatic control		17 (3)
Oncology biomarkers		14 (3)
S100B protein		14 (3)
Urinary cells		13 (3)

Abbreviations: crp: C-reactive protein; BNP: brain natriuretic peptide; hCG: humain chorionic gonadotrophin.

**Table 3 jpm-12-00106-t003:** Cardiac biomarkers and demographic characteristics.

Demographic Characteristics	Biomarker		*p* *	OR **	*p* **
	Troponin quoted (N = 257)	Troponin non-quoted (N = 251)			
Age (years)	37 ± 10	38 ± 11	0.41	1.01 (0.98–1.03)	0.71
MD	199 (77)	208 (83)		Réf.	-
Resident	58 (23)	43 (17)		0.82 (0.48–1.42)	0.48
EM	79 (31)	83 (33)		Réf.	-
GP	152 (59)	72 (29)		3.58 (1.82–7.03)	<0.0001 ***
Other	26 (10)	96 (38)		0.36 (0.19–0.68)	<0.0001 ***
Public	165 (64)	145 (68)		Réf.	-
Private	92 (36)	106 (42)		1.55 (0.86–2.79)	0.14
France	192 (75)	184 (73)		Réf.	-
Europe (excluding France)	20 (8)	27 (11)		0.95 (0.45–2.02)	0.40
USA	33 (13)	25 (10)		0.63 (0.32–1.25)	0.55
Other	12 (5)	15 (6)		0.50 (0.21–1.19)	0.24
	D-dimer quoted (N = 146)	D-dimer non-quoted (N = 362)			
Age (years)	37 ± 10	48 ± 11	0.24	1.02 (0.99–1.05)	0.15
MD	113 (77)	294 (81)		Réf.	–
Resident	33 (23)	68 (19)		0.76 (0.43–1.35)	0.35
EM	61 (42)	101 (28)		Réf.	–
GP	73 (50)	151 (42)		1.50 (0.71–3.15)	0.31
other	12 (8)	110 (30)		0.24 (0.11–0.53)	<0.0001 ***
Public	80 (55)	230 (64)		Réf.	–
Private or mixed	66 (45)	132 (36)		1.63 (0.83–3.21)	0.16
France	123 (84)	253 (70)		Réf.	–
Europe (excluding France)	7 (5)	40 (11)		0.43 (0.17–1.09)	0.18
USA	11 (8)	47 (13)		0.32 (0.14–0.70)	0.001
Other	5 (3)	22 (6)		0.38 (0.14–1.08)	0.28
	BNP/NT-proBNP quoted (N = 67)	BNP/NT-proBNP non-quoted (N = 441)			
Age (years)	39 +/− 11	38 +/− 10	0.30	1.02 (0.99–1.05)	0.26
MD	51 (76)	356 (81)		Réf.	-
Resident	16 (24)	85 (19)		0.63 (0.31–1.3)	0.21
EM	18 (27)	144 (33)		Réf.	-
GP	31 (51)	190 (43)		2 (0.78–5.13)	0.09
other	15 (22)	107 (24)		1.1 (0.46–2.66)	0.47
Public	41 (61)	269 (61)		Réf.	-
Private	26 (39)	172 (39)		1.61 (0.73–3.56)	0.24
France	46 (69)	330 (75)		Réf.	-
Europe (excluding France)	11 (16)	36 (8)		1.89 (0.82–4.4)	0.07
USA	6 (9)	52 (12)		0.59 (0.22–1.63)	0.14
Other	4 (6)	23 (5)		1.03 (0.33–3.20)	0.98

* Student’s *t*-test or Fisher’s exact test; ** multivariate logistic regression, *** *p* < 0.05. Abbreviations: MD, medical doctor; EM, emergency physician; GP, general practice.

## Data Availability

The data presented in this study are available on request from the corresponding author. The data are not publicly available due to a reuse in progress for another analysis.

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
