# Peer review of "Relevant Biomarkers in Medical Practices: An Analysis of the Needs Addressed by an International Survey"

_jpm, 2022, doi:10.3390/jpm12010106_

Round 1

Reviewer 1 Report

In the submitted manuscript, the authors attempted to answer the urgency of biosensors to the current biomarkers in clinics. They distributed questionnaires to 508 physicians in this survey globally. They found faster diagnosis is needed for certain biomarkers. I found this work is interesting and the research angle is unique. I support its publication if the author may address my concerns.

First, the questionnaire should be listed as supporting information with this work as it is the key part to assess the validity of this work. Whether the survey is scientifically designed depends on it.

Second, the current title of this work dose not reflect the content. It looks like a review article instead of original research.

Author Response

Dear reviewer,

We sincerely thank you for your interest in our manuscript.

1/ The questionnaire is already in the article, in Appendix A. However, if the reviewer wishes and in agreement with the editor, we will move it to the text.

2/ We are in complete agreement with the point you raise concerning the title of the article and we propose this new title: "Relevant biomarkers in medical practices: an analysis of the needs addressed by an international survey"

Have a nice end of the year.

Best Regards,

Reviewer 2 Report

In the paper, the authors  have collected  responses from an international panel of physicians on their need for biomarkers, in order to convey the improvement in outcome to be made possible by the potential development of faster diagnostic tests for these biomarkers. The response from 508 physicians identified as the main biomarkers of interest: troponin, C-reactive protein , d-dimer, and brain natriuretic peptide. Overall, the author conclude that that there is a strong need to speed up care decisions and the flow of information by quickly obtaining biology reports.

The paper is interesting and the methods appropriate.

The paper should go a thorough grammar- and spell-check for better readability (for instance: “Fischer”should be Fisher; the second sentence in the abstract is incomplete…)

Author Response

Dear reviewer,

We sincerely thank you for your interest in our manuscript.

English has been proofread by a native English speaker for this new submission.

Have a nice end of the year.

Best Regards,